# The Role of Age, Neutrophil Infiltration and Antibiotics Timing in the Severity of *Streptococcus pneumoniae* Pneumonia. Insights from a Multi-Level Mathematical Model Approach

**DOI:** 10.3390/ijms21228428

**Published:** 2020-11-10

**Authors:** Guido Santos, Julio Vera

**Affiliations:** 1Laboratory of Systems Tumor Immunology, Department of Dermatology, Universitätsklinikum Erlangen and Faculty of Medicine, Friedrich-Alexander University Erlangen-Nürnberg, 91052 Erlangen, Germany; gsantos@ull.edu.es; 2Departamento de Bioquímica, Universidad de La Laguna, Microbiología, Biología Celular y Genética, 38200 San Cristóbal de La Laguna, Spain

**Keywords:** multi-level, mathematical model, *Streptococcus pneumoniae*, pneumonia, macrophages, neutrophils, monocytes, inflammation

## Abstract

Bacterial pneumonia is one of the most prevalent infectious diseases and has high mortality in sensitive patients (children, elderly and immunocompromised). Although an infection, the disease alters the alveolar epithelium homeostasis and hinders normal breathing, often with fatal consequences. A special case is hospitalized aged patients, which present a high risk of infection and death because of the community acquired version of the *Streptococcus pneumoniae* pneumonia. There is evidence that early antibiotics treatment decreases the inflammatory response during pneumonia. Here, we investigate mechanistically this strategy using a multi-level mathematical model, which describes the 24 first hours after infection of a single alveolus from the key signaling networks behind activation of the epithelium to the dynamics of the local immune response. With the model, we simulated pneumonia in aged and young patients subjected to different antibiotics timing. The results show that providing antibiotics to elderly patients 8 h in advance compared to young patients restores in aged individuals the effective response seen in young ones. This result suggests the use of early, probably prophylactic, antibiotics treatment in aged hospitalized people with high risk of pneumonia.

## 1. Introduction

Pneumonia is the cause of 20% of deaths in children worldwide [1], with *Streptococcus pneumoniae* infection being the most common cause [2]. Pneumonia also affects other risk population groups like elderly people or hospitalized patients [3,4]. Local and systemic inflammation in the course of the infection is the main risk factor in these patients [5], which makes controlling inflammation in the early phases of lung infection the most promising strategy to prevent mortality in high-risk groups [6,7].

During the first hours of the infection, passive and innate immunity lead the response against bacteria and inflammation. Resident alveolar macrophages keep surveillance of agents entering the alveolus together with the epithelial cells. These cells activate inflammation together with other immune cells recruited during the activation process. In the early infection, monocytes act as phagocytes supporting alveolar macrophages and neutrophils to produce an acute inflammatory and bactericide response. The immune system produces and secretes several cytokines to facilitate cell recruitment and modulate inflammatory response. Upon bacteria sensing, the epithelial cells of the alveolus synthesize and secrete the chemokine MCP-1, which controls phagocyte recruitment, and CXCL5, which recruits neutrophils [8,9]. Further, phagocytes produce IL-8, a cytokine that modulates the recruitment of neutrophils [10]. When the infection is not quickly controlled by the innate immune response, the adaptive immunity produces a long term, specific inflammatory response against the pathogen. However, to mount this adaptive immunity can take a period of several days and this delay induces in many cases a systemic, overwhelming inflammation, which makes pneumonia reach a point of no return and drastically increases clinical risk. Furthermore, the acute immune response triggered by *S. pneumoniae* inside the alveoli alters the tissue homeostasis and promotes liquid accumulation inside the alveolar tissue, thereby hindering normal breathing. Alveolar tissue regeneration to restore the damaged epithelium is activated only when bacterial infection is controlled. However, since the process can get delayed in time, acute, long-lasting lung infection jeopardizes tissue recovery and patient survival.

*S. pneumoniae* alveolar infection is mostly controlled with antibiotics. Complications due to the bacterial infection (for example, septicemia or secondary infections) are less frequent in patients than those linked to lung inflammation. Although vaccines against *S. pneumoniae* are available, their efficiency is low in high-risk populations due to the limited immune response triggered by the vaccine in these patients [11,12]. Thus, the best strategy to control pneumonia in a high-risk patient is reducing the bacterial load during the early phases of infection before producing an overwhelming immune response. Early or prophylactic antibiotics treatment could reduce the inflammatory response triggered in the lungs by *S. pneumoniae* during pneumonia. We talk about early antibiotics treatment when the decision criteria are based on the individual patient prognosis instead of on general clinical recommendations [13]. This strategy could benefit sensitive patients exposed to events with high risk of pneumonia, like immunocompromised patients hospitalized due to other severe conditions. Specifically, aged hospitalized patients under ventilation-based treatment present a high incidence of pneumonia [13]. Early antibiotics treatment could be optimal in these conditions.

Mathematical model simulations allow testing multiple hypotheses in order to select optimal treatments for their validation in vivo [14,15]. Calibrated computational models combined with experimental data can be used to dissect the regulatory pathways controlling immune cells or bacteria in the course of infection [16,17], to find new biomarkers for disease prognosis [18], to analyze the feasibility of conventional or personalized treatments, to detect new drug targets [19,20], or to investigate the ecological interactions between bacterial species during respiratory tract infection [21,22,23]. In line with this approach, in this paper we derived and characterized with available experimental data a multi-level mathematical model able to simulate the first 24 h of *S. pneumoniae* infection inside a single lung alveolus. The objective of the present work is to use this computational model to investigate in silico the relevance of early antibiotics treatment in elderly patients.

## 2. Results

We proposed a model able to simulate the first 24 h after bacterial lung alveolus infection. This model aims to predict the most relevant cellular and molecular mechanisms driving *S. pneumoniae* infection during the very first stages of pneumonia, which cannot be entirely tracked by experimentation. The model considers both the barrier defenses and innate immune system against *S. pneumoniae,* while the adaptive immune system response is assumed to play a relevant role in later phases of the infection. To cope with all the relevant spatiotemporal features, we derived and characterized a hybrid, multi-level mathematical model, which combines agent-based, partial and ordinary differential equations.

### 2.1. Biological Scenarios Modelled

In this work, we analyze the effect of the neutrophil load on alveolar infection. Neutrophils play a major role during the early phases of pneumonia that are controlled by the innate response [24]. It has been observed that there is a reduction in neutrophil load in aged mice [25]. Here, we hypothesized that a decrease in neutrophil levels could explain the sensitivity to pneumococcal infection in elderly patients. We implemented this observation in our model by decreasing the recruitment rate of neutrophils during infection. Two groups of solutions were created based on this strategy, a group named “nominal”, which accounts for the simulations obtained with the nominal parameter values for neutrophil recruitment as in Appendix A, and a group named “aged” which accounts for simulations in which the recruitment of neutrophils was reduced in a magnitude similar to that observed in Chen et al., 2014 [25].

We defined the nominal solution of the model as the starting point of the perturbation analysis, which represents the average condition that happens during an infection of a lung alveolus with *S. pneumoniae* in a young, healthy individual (see Material and Methods for details). The nominal condition was obtained by combining quantitative information from the literature and calibration of given model parameters using in vivo [26] experimental data (See Appendix A). Taken together, the nominal solution is calibrated to simulate young specimens [26]. Model simulations accounting for the nominal solution are shown in Figure 1. Because the model presents stochasticity in the movement of the agents representing bacteria and immune cells, in the figure we display ten model simulations with the same parameter values and initial conditions. Figure 1A displays the evolution over time of the populations for bacteria, monocytes, macrophages and neutrophils as predicted by the model. The four variables are normalized with respect to the initial amount of macrophages in the alveolus, which allows for interpreting the amount of bacteria as MOI with respect to the number of macrophages in resting condition. Figure 1A indicates that in nominal conditions bacteria grow during the first 20 h; after that point, bacteria levels decrease as a consequence of the infiltration of monocytes and neutrophils. In the model simulations, cell recruitment presents a monotonic increase corresponding to the acute inflammatory profile observed days later in the animal model [27]. In Figure 1B chemokines show a slow growing accumulative pattern due to the progress of the inflammatory process. Pneumolysin presents a peak of production before the 15 h mark, which is according to the experimental evidence [28].

In our simulations we distinguish between resident alveolar macrophages and monocytes-derived phagocytes. In physiological conditions, a single alveolar macrophage patrols several interconnected alveoli, which leads to inclusion of a single macrophage in the alveolus at the beginning of the simulation [29]. Interestingly, in the simulation the resident alveolar macrophage is depleted in the course of the infection and during the first 10 h. This is due to the over-accumulation of bacteria in the alveolus, an event that triggers programmed cell death in macrophages [30].

Next to the nominal solution, we included a second scenario in our model to simulate the effect of aging in the fitness of the innate immune system [31,32]. To this end, we modified the value of the model parameter that accounts for neutrophil recruitment. The experimental results in Chen et al., 2014 [25] suggest that in the context of lung infection a relevant difference between young and aged individuals is the levels and the recruitment rate of neutrophils, which can be reduced up to 80 % in aged mice compared to young ones. Model simulations accounting for the impaired neutrophil recruitment are compared with the nominal solution for key model variables in Figure 1. In Figure 1 we can see that in the nominal condition the bacterial population grows in the range of tens to hundreds as a consequence of the proper innate immune response. In contrast and under aged-related impairment of neutrophil recruitment, the model predicts up to thousands of bacteria by the end of the simulated period. According to the simulations, up to few tens of neutrophils are recruited to the alveolus in the nominal solution, while a maximum of three neutrophils reach the alveolus in the aged scenario.

### 2.2. The Effect of Antibiotics Administration Timing in Early Lung Infection for Aged Individuals with Impaired Neutrophils Recruitment

In order to investigate the dynamics of lung alveoli infection by *S. pneumoniae* under antibiotic administration, we simulated the continuous administration of penicillin as described in the Material and Methods section. Next, we performed perturbation simulations in which we modified the time after infection in which antibiotic administration starts in both the young and aged scenarios in conjunction with changes in the initial amount of bacteria infecting the alveolus (Figure 2).

The initial amount of bacteria infecting a single alveolus depends on the distribution of *S. pneumoniae* on the alveolar tissue from the higher airways. In our simulations, we used this variable to get a profile of the infection in the whole lung. With high values of initial bacteremia, we simulated alveoli that are close to the infection loci in the lungs, while with low values we account for alveoli far from the infection loci. The bacteremia values we tested increased gradually from 11 to 151 bacteria, 10 bacteria in each iteration.

The second model variable that we perturbed during the simulations stands for the time after infection in which antibiotic administration starts. In the model we assumed continuous perfusion of penicillin, which corresponds to a clinical setup of intravenous antibiotics administration [33]. We used the equivalent to a penicillin dose used in experiments [26]. In the simulation, we iteratively modified the time from the infection initiation until the antibiotics reach the bloodstream (τ). The values ranged from 0 to 20 h after infection (0, 4, 8, 12, 16 and 20 h).

In order to quantify the dynamics of the infection during the simulation, we computed three different magnitudes that account for bacteria dynamics in the course of the infection inside the alveolus (namely, *R*_0_, *Dif* and *Inc*). Each of these metrics provides a different view of the bacteremia dynamics during infection. *R*_0_ provides an estimate of the potential dispersion of the infection through the lungs, *Dif* is the average of the increase rate of the infection and *Inc* measures the bacteremia increase between the initial time and 24 h later. In Figure 2 we display the results of the simulations for the two scenarios of neutrophil recruitment, as well as different antibiotics timing and initial bacteremia investigated. Similar patterns were observed in all three magnitudes computed. Interestingly, the simulations indicate a significant difference between the young and aged scenarios regarding the effect of the delay in the administration of antibiotics. On average, the dynamics of the infection is delayed approximately 8 h in young versus aged scenarios.

An interesting pattern is observed in the normalized integral of bacteria and the normalized increment of bacteria in the aged situation for 111 initial bacteria in the antibiotic regimen (τ) of 20 h. In this scenario these two measurements reach their maximum, even above the values they get for the higher number of initial bacteria. Our analysis suggests that, since this effect is only observed in the two measurements that normalize by the initial number of bacteria, it is linked to the interplay between initial and maximum bacterial load in the alveolus in this scenarios and metrics.

Figure 3 further elaborates on the effect of τ and age on bacteremia. The figure shows the expected pattern of decrease on the amount of bacteria after antibiotics administration in all the scenarios simulated. However, only in the simulation for young individuals with τ = 12 h (Figure 3A, middle) bacteria levels at the end of the simulation are comparable to the initial state and hence a reflection of successful depletion of the infection. In contrast, in aged individuals and τ = 12 h the bacteremia is decreasing from its maximum levels at the end of the simulation, but still stays very high (Figure 3B, middle). To control the bacteremia at the end of the simulation in aged individuals, one has to administer the antibiotics at τ = 4 h (Figure 3 panel B, left).

## 3. Material and Methods

### 3.1. Model Description

The mathematical model presented accounts for the first 24 h of infection of a lung alveolus by *S. pneumoniae*. In the model, the lung alveolus is represented as a truncated frequency two icosahedron. Bacteria and immune cells move through the inner surface of the icosahedron and chemical signals like chemokines and cytokines diffused on the inner surface.

Our model is built on previous models developed by our group and others describing the signaling pathways and the cell-to-cell interactions between host and bacteria triggered during early time after infection [27,34]. This model was calibrated using data from *S. pneumoniae* experiments in vitro and in vivo. In the model, epithelial cells can recognize and react to bacteria producing chemokines (MCP-1) that attract the resident macrophages to the point of infection. The model also considers the dynamics of the alveolar lining liquid that can wash out bacteria from the alveolus. The model further simulates the recruitment of immune cells to the infection site. We here focus on the innate immune system response, which precedes the adaptive response by one or more days. In our model, monocytes appear to support the function of resident macrophages, while neutrophils act as the main destructive agent against the increasing infection. Both cell types are recruited following chemical signals secreted by both local macrophages and epithelial cells in response to bacteria. Macrophages produce interleukin 8 (IL-8), which attracts neutrophils from the bloodstream to the alveolar tissue, while the epithelial cells chemokine MCP-1 recruit monocytes. Another signal coming from the epithelial cells included in our model is CXCL5, which also recruits neutrophils. In the model we integrated in each epithelial, monocyte and macrophage cell agent, an NF-κB-centered intracellular network is responsible for the triggering of cytokines and chemokines secretion. In case of the IL-8 produced by macrophages and monocytes in response to bacteria, we assumed for IL-8 the same model parameter values than for MCP-1 based on the fact that both ligands show similar expression profiles in relevant in vitro experiments [35]. Further, in the model we assumed that the CXCL5 secretion is triggered in the epithelial cells through the same NF-κB-centered network that triggers MCP-1. Finally, we calculated the diffusion constants for each cytokine and chemokine in the alveolus using an empirical estimate based on the molecular weight of macromolecules [36].

We also included the production of the virulence factor pneumolysin by *S. pneumoniae*, primarily responsible for the bacteria-associated tissue damage. In order to fit the model to experimental observations concerning pneumolysin [28], we set two production phases: a first one with high pneumolysin production and after 15 h a second one with low production. An important feature of *S. pneumoniae* lung infection is the existence of immunomodulatory effects promoted by proteins and other compounds of the bacteria surface and capsule [37,38,39]. In our model, we assume that these immunomodulatory processes are negligible during the very initial phase of the infection here modelled, in which bacteria are adapting to the new environment and the rate of capsule proteins and compounds synthesis is low (see discussion about this issue in Santos et al. 2018) [27].

Our model is a multi-level hybrid model, which utilizes agent-based modelling to represent the movement of cells in the tissue, partial differential equations for the gradients of cytokines and ordinary differential equations for the intracellular pathways. The agents and processes included in the model are described in Figure 4A. In the model simulations, bacteria and immune cells move through the inner surface of the icosahedron, while chemokines diffuse, thereby mimicking the real environment of an alveolus infection (see Figure 4B and video of the simulation of the model with nominal conditions in this link). All processes and parameters included in the model are included in Appendix A. This table also includes the values of the parameters in the nominal solution and the references from which the values were taken or calculated. The nominal solution represents a condition of an animal model of pneumonia (mouse) subjected to an infection by *S. pneumoniae* (MOI = 10 bacteria per resident macrophage). This condition produces acute pneumonia that a subset of the exposed animals is able to overcome [26].

### 3.2. Simulations

#### 3.2.1. Handling of Stochasticity

The model presents a stochastic behavior on the movement of cells through the inner surface of the alveolus. Due to its stochasticity, the same set of parameters can produce different outcomes every time the model is executed. In order to deal with this stochastic process, we simulated every solution 10 times. As we can see in Figure 1, all the 10 simulations of the nominal solution follow the same qualitative behavior. We set the seed for the simulations in order to make a proper comparison between the group “nominal” and “aged.” In order to consider this stochastic variability, we represented the output of the model for any set of parameters tested as the average of 10 simulations. Using this approach, we investigated the evolution of the pneumococcal infection under different biological scenarios.

#### 3.2.2. Modelling of Antibiotics Administration

We used data from in vitro experiments with penicillin to calculate the clearance rate of pneumococci [24] and we introduced this parameter in the model. We performed a scenario of antibiotics treatment in which one provides a constant supply of penicillin at a given time point after the onset of the infection. This situation reproduces a clinical scenario of intravenous antibiotics treatment. In the simulations, we used the time before initiating antibiotics as a tunable parameter. This parameter was modified between 0 and 20 h after onset of infection in intervals of four hours (τ = {0, 4, 8, 12, 16, 20} hours).

#### 3.2.3. Modelling Variability in Bacterial Load

Another parameter that presents variability in vivo is the initial load of pneumococci that initiate the infection inside the alveolus. To account for this, we allowed values of initial bacteria from 11 to 151 per alveolus in each group (11, 21, 31…, and 151). Eleven bacteria per alveolus represent the nominal situation of a standard primary infection, calculated from mice experiments data (see Figure 1 and Appendix A). We used higher amounts to simulate patients infected in highly infectious locations like hospitals, in which we assume higher initial bacteremia are feasible.

#### 3.2.4. Solutions

Combining modifications in both timing for antibiotics administration and bacterial load, we obtain 48 solutions inside each group (*nominal* and *aged*). Each solution is represented by the average of 10 simulations with the same parameter values, which produced a total amount of 480 simulations inside each group (960 simulations in total). Each simulation is run from 0 to 24 h after infection onset.

#### 3.2.5. Metrics for Infection Progression

To compare the evolution of the disease between the two groups, this requires considering the variability that the simulations present in time. In order to deal with this, we computed three measurements accounting for the infection progression defined as the following.

*R*_0_: a magnitude similar to the basic reproduction number used in epidemiology provides an estimate of the potential dispersion of the infection through the lungs. Similar to the epidemiologic *R*_0_ parameter, a value lower than 1 corresponds to a state in which the epidemic decreases. In order to get a parallel estimate for the evolution of infection in the lungs, we compare the total cumulative amount of bacteria present in the alveolus respecting a state in which the initial amount of bacteria stays unchanged during the infection. We express this comparison by the following ratio:(1)$$R_{0}=\frac{\int Bacteria\cdot dTime}{Bacteria\left(Time=0h\right)\cdot Timespan}$$

Values of *R*_0_ higher than 1 indicate that the course of infection has produced more bacteria than the ones existing at the moment of infection, time = 0. Any alveolus acts as an infective point for other neighbor alveoli by spreading bacteria from it. Thus, *R*_0_ can be interpreted as an estimate of the bacterial spread through the lungs. Values higher than 1 indicate that the total number of bacteria present in the alveolus at any time is higher than the expected number of bacteria in the alveolus if the initial bacteria remains unaltered inside it.

*Dif*: average of the increase rate of the bacteremia. It is a metric that considers the increments on bacteremia during the course of infection and is calculated as:(2)$$Dif=\sum \frac{dBacteria}{dTime}$$

This measurement is an estimate of the average bacteremia increase during the timespan. The higher the value, the higher the growth rate of bacteria in vivo. If *Dif* is higher than 0, it indicates a net growth of bacteria during the time of infection, while values lower than 0 indicate a net decrease in the bacteria population.

*Inc*: bacteremia at the last stage of infection. It measures the bacteremia increase between the initial time and 24 h, that is, the end point of the simulation. In contrast with other parameters, it only considers the initial and last state of infection:(3)$$Inc=\frac{Bacteria\left(Time=24h\right)-Bacteria\left(Time=0h\right)}{Bacteria\left(Time=0h\right)}$$

This measurement ignores the dynamics during the evolution of the disease and considers only the last stage of the infection. Values higher than 1 indicate that there has been an increase in the amount of bacteria at the end of the infection respecting the initial state, while values lower than 1 show that the amount of bacteria has decreased with respect to the beginning.

### 3.3. Running Environment for Simulations and Availability of the Source Code

The model was implemented in Matlab 2012b (Natick, MA, USA). The simulations were performed in a cluster with 32 CPUs and 128 GB RAM memory. Each solution took on average 10 h. The files containing model equations and rules, seeds for stochastic simulations and simulations are provided here.

## 4. Discussion

Controlling bacterial pneumonia in patients that have reached the acute phase is complicated due to the involvement of an uncontrolled inflammatory response. This severity occurs often in immunocompromised patients like those from the aged population. Moreover, aged patients in hospitals are exposed to nosocomial infections like *Streptococcus pneumonia*, among others. There is a need for finding early diagnostics and therapeutic strategies able to reduce the risk of these patients entering the acute phase of pneumonia.

To this end, according to our simulations, during the first hours of infection the activation and recruitment of neutrophils is one of the main agents involved in this overwhelming inflammation in aged individuals. There is experimental evidence suggesting an improvement in patients by acting on the antibiotics timing [40], but to our knowledge the effect on early antibiotic administration in the first stage of the infection has not been evaluated on aged patients.

To this end, we utilized a multi-level mathematical model to predict the effect of early antibiotics treatment on an elderly clinical model of the disease. We base our predictions on a multi-level mathematical model calibrated with a clinically accepted mouse model of CAP pneumonia [26]. Our model has been calibrated using data from experiments with *S. pneumoniae*. We have focused on *S. pneumoniae* because it is the most prevalent bacterial lung infection. However, the current implementation of our model could be used with little changes to simulate the infection dynamics associated with similar bacterial species.

To reproduce the situation of the elderly model we modified the parameters to fit the immunological differences observed on aged mice with pneumonia [25]. This provided us with two different instances of the model that reproduce the first hours of *S. pneumoniae* infection of an alveolar tissue in both aged and young individuals (see Figure 1). We computed three different bacteremia metrics to compare the simulated infection profiles for young and aged individuals, the results show slight differences in the values for the integral metrics (*R*_0_, *Dif*) compared to the one accounting for the instant bacteremia values at the end of the simulation (*Inc*). This suggests that the scenarios analyzed cover the saturation on bacterial growth inside the alveolus for a high amount of initial bacteria. Our simulations also suggest substantial differences in the dynamics of the infection between the two clinical scenarios. Younger individuals respond better to the alveolar infection producing far better measurements concerning bacteremia.

The maximal similarity between the simulations for young and aged individuals occurs for very early administration of antibiotics in aged individuals (τ = 4 h) compared to mid-time administration in young individuals (τ = 12 h). Although younger individuals respond better to the very early administration of antibiotics, the improvement can be observed in both age groups (Figure 3). These results suggest that antibiotic timing has the capacity to counterbalance the impairment of the immune response linked to the smaller early recruitment of neutrophils to the infection site. We hypothesize that early, even prophylactic antibiotics administration can reduce the risk of acute pneumonia for elderly patients, especially during hospitalization [41]. The underlying idea of the hypothesis is that by drastically reducing the levels of bacteria in the very early phase of the infection, we avoid the mid-term activation of local and systemic cytokine and chemokine-mediated autocrine loops. These loops, when activated, could render the inflammation linked to the infection unresolved, even under (late) administration of antibiotics (Figure 5A,B).

In this sense, profiling blood levels of neutrophils could be a surrogate to decide for preventive of early antibiotics treatment in different patients. Further clinical assays based on these results could identify neutrophils as biomarkers to identify sensitive patients that would require early antibiotics after symptoms first appear during hospitalization. The main limitation for an early antibiotic treatment is the delay observed from diagnosis [42]. Our results indicate that investing in accelerating diagnosis for elderly patients in hospitals is a promising strategy for pneumonia that could compensate for the age factor in these patients. Based on our results we proposed a decision tree for a personalized treatment of pneumonia (Figure 5C). This decision tree could be validated by clinical studies with patients under different conditions. The proposed protocol would be applied as follows: patients with a planned (surgery, dialysis, chemotherapy) or unplanned (trauma) hospitalization involving a pneumonia high-risk intervention (intubation, tracheostomy, etc.) would be split into high-risk and low-risk patients based on age and other immunological conditions. Neutrophils levels would be quantified in high-risk patients to further determine pneumonia risk. Finally, patients categorized as high-risk based on both age/immune state and low neutrophil count would receive prophylactic antibiotics in order to reduce their risk of bacterial pneumonia during or post-intervention.

Our multi-level model only reproduces the evolution of the infection in a single alveolus. To generate with the model a spectrum of alveoli infection based on the level of exposure to bacteria, we systematically modified the initial amount of bacteria infecting the alveolus and simulated the dynamics of infection. The underlying assumption is that the only difference between the lung alveoli at the initial point of infection is the amount of bacteria that they receive during the colonization of the alveolar tissues from the upper airways. Thus, we think our systematic simulations in which we vary bacteremia give a picture of the initial phases at the whole lung scale. However, the model cannot reflect the role played by other immune cell populations and branches of the immune response in the later phases of infection, but also cannot account for systemic processes induced by the parallel infection of a large fraction of the alveoli in the lung. These are features that require an extended version of the model with more cell types, cytokines, spatial processes and interlinked alveoli.

The mouse experimental model used to build the model in the present studio can differ from community-acquired pneumonia (CAP) in humans because mice are inoculated in the experiments with a much higher amount of bacteria to ensure infection. However, this experimental model of pneumonia is considered a valid alternative to study the evolution of pneumonia in humans [26]. Also, we expect that the bacterial local infection inside at the alveoli scale will not differ between both situations. Instead, the initial distribution of bacteria in the whole lungs can represent the main contrast between actual CAP and the experimental model.

## Figures and Tables

**Figure 1 ijms-21-08428-f001:**
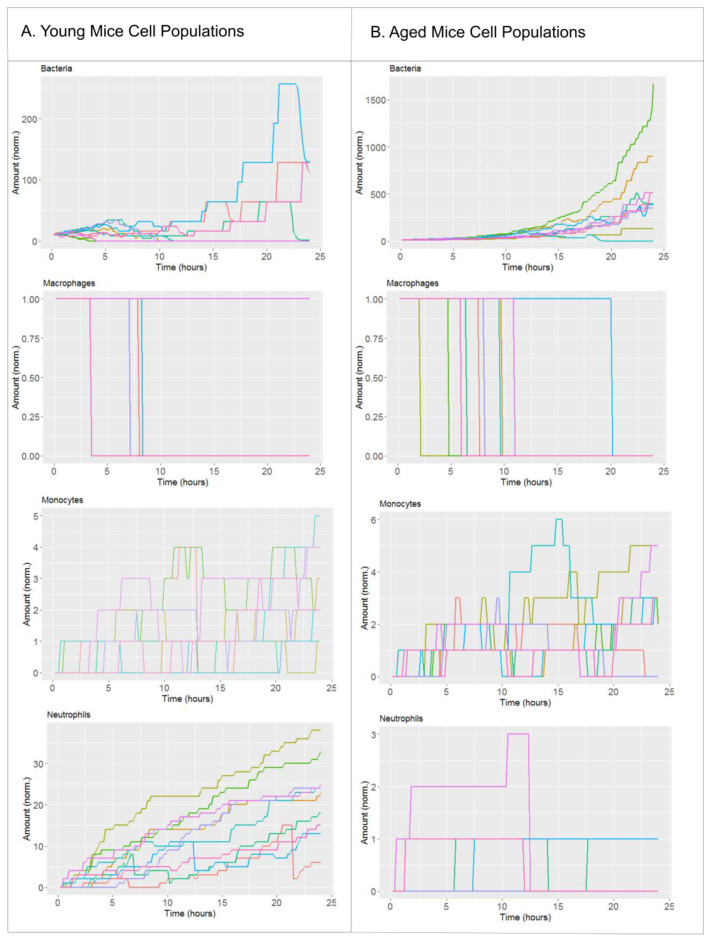
Simulation of the nominal solution. (**A**) Cellular components of the nominal solution of the model (young mice). (**B**) Cytokines and virulence factors of the nominal solution of the model (young mice). (**C**) Cellular components of the simulations of aged mice. (**D**) Cytokines and virulence factors of the simulations of aged mice. We display 10 simulations for the parameter set of the nominal solution (Appendix A). Each solution is represented by a different color.

**Figure 2 ijms-21-08428-f002:**
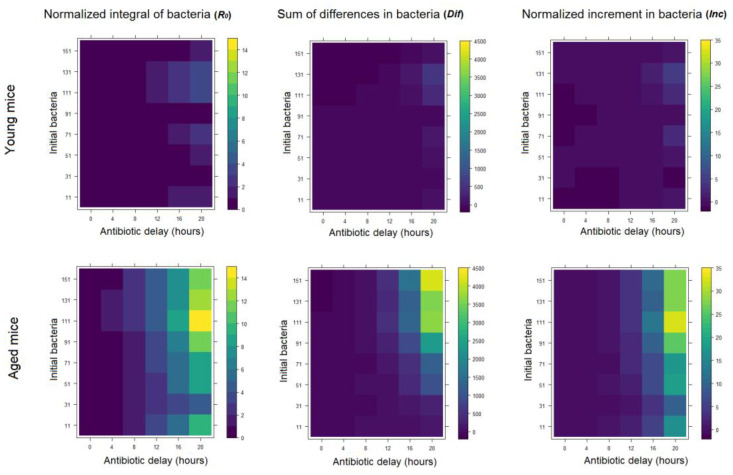
Three different bacterial indicators to compare nominal versus aged conditions in the model for different initial bacteria and antibiotic delay. The first column considers the measurement *R*_0_ (Basic reproduction number: an estimate of the potential dispersion of the infection through the lungs), the second column uses the measurement *Dif* (diference: average of the increase rate of the bacteremia) and the third one the *Inc* (*increment*: bacteremia at the last stage of infection).

**Figure 3 ijms-21-08428-f003:**
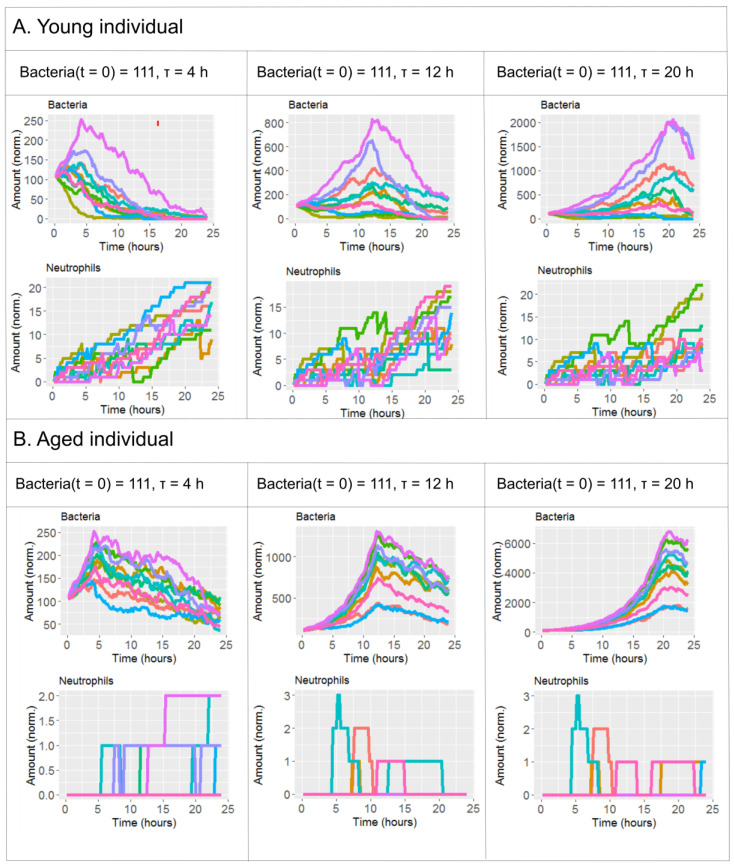
(**A**) Simulations of four different conditions modifying the antibiotic time regimen (τ) for young individuals. (**B**) Simulations of four different conditions modifying the antibiotic time regimen (τ) for aged individuals. Additional variables are shown in Appendix A.

**Figure 4 ijms-21-08428-f004:**
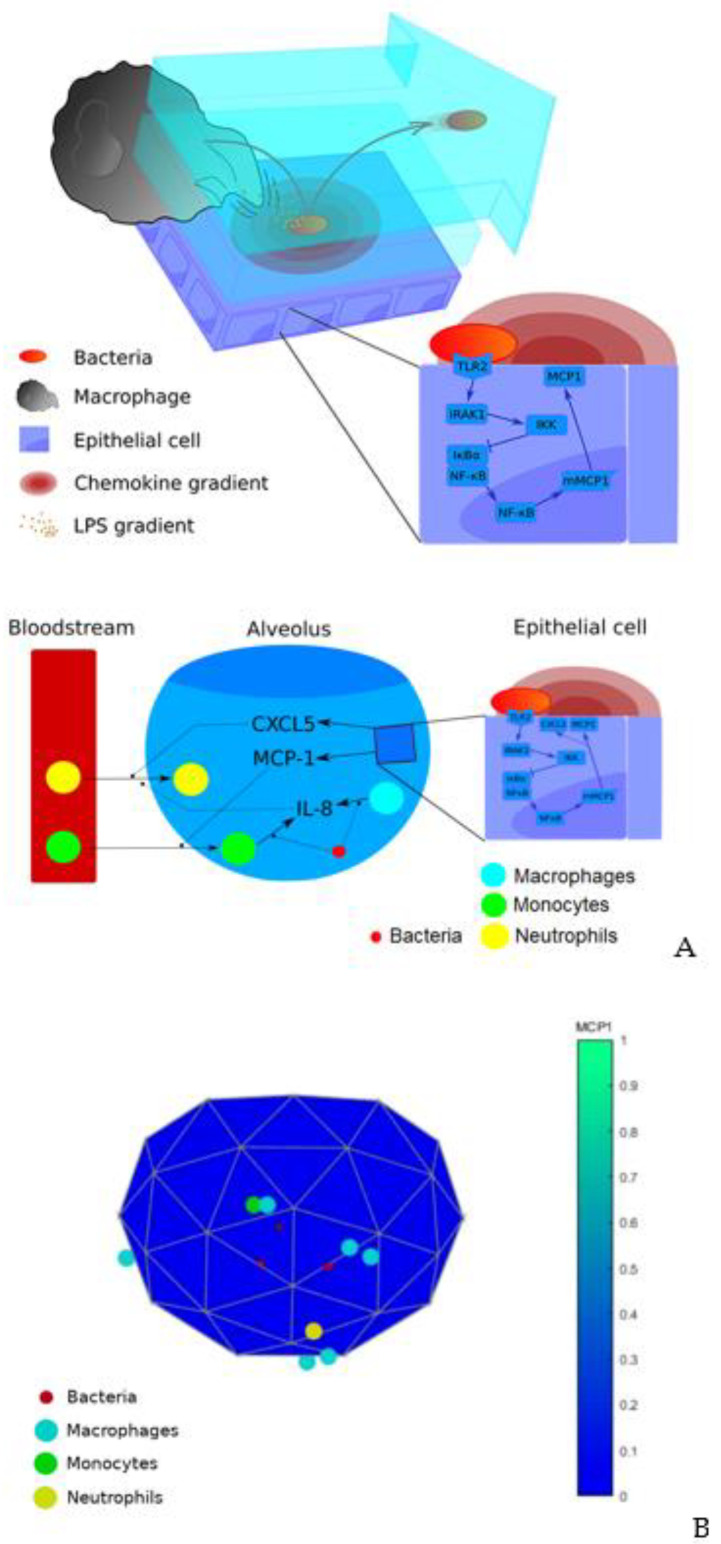
(**A**) Diagram of previous model. (**B**) Frame of the simulation of the extended model. The color bar represents the average concentration of the chemokine MCP1 between all epithelial cells and normalized to a sufficient high value to make it visible during the whole simulation (2.4·10^−3^ ng/mL).

**Figure 5 ijms-21-08428-f005:**
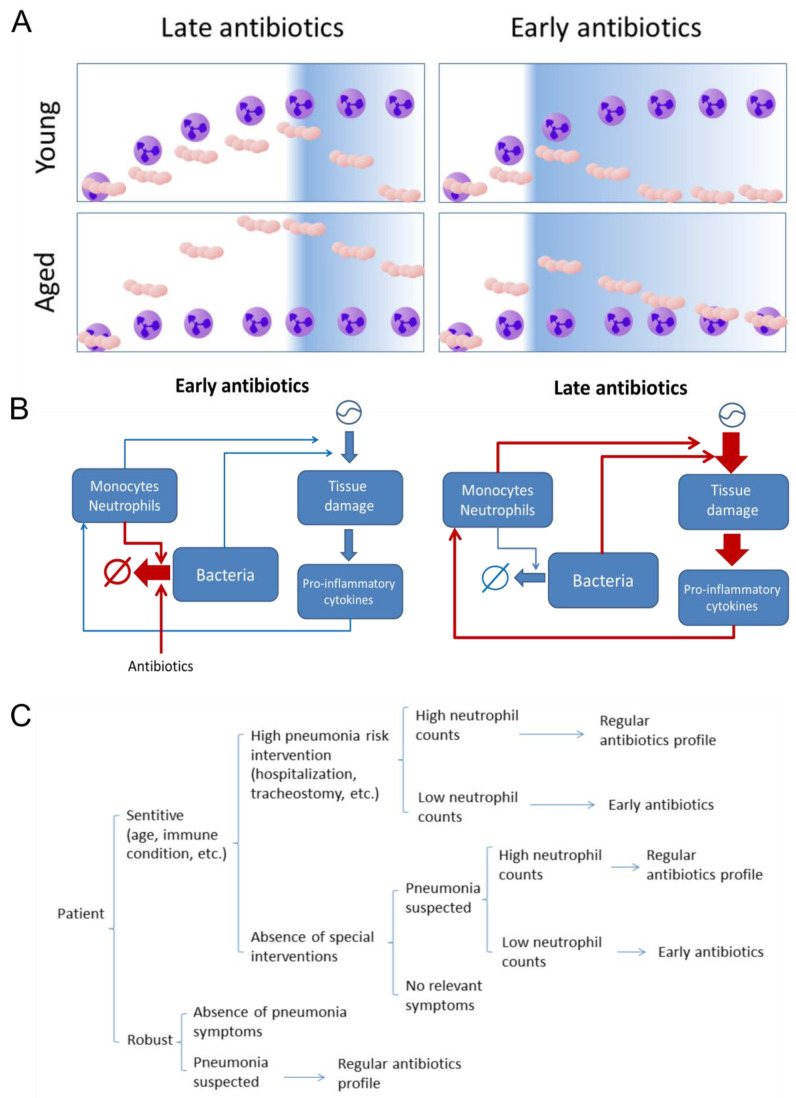
(**A**) Graphical representation of the effect of early antibiotic administration on bacteremia under the two different age scenarios (purple cell: neutrophils, beige cocci: *S. pneumoniae*). (**B**) The hypotheses derived from the simulations is that under late antibiotics administration, sustained bacteremia levels induce concomitant long-lasting tissue damage and inflammation, which activates later cytokine-mediated feedback loops promoting unresolved inflammation (not shown in the simulations). Under early antibiotics administration, the bacteremia and tissue damage never reach the levels and duration necessary to trigger the sustained feedback activation. (**C**). Sketch of a possible decision tree for a personalized treatment of pneumonia based on neutrophil count.

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
