# Peer review of "The Role of Age, Neutrophil Infiltration and Antibiotics Timing in the Severity of *Streptococcus pneumoniae* Pneumonia. Insights from a Multi-Level Mathematical Model Approach"

_ijms, 2020, doi:10.3390/ijms21228428_

Round 1

Reviewer 1 Report

The current article uses mathematical modeling to determine the role of age on neutrophil recruitment and antibiotic treatment on pneumococcal pneumonia. The authors use a previous model of pneumococcal alveolar lung infection that mimics a single alveoli and host cytokine and innate immune response. Using this model the authors address the growth of S. pneumoniae in relation to age. The defining characteristic of age was reduced neutrophil recruitment, which mediated pneumococcal clearance. Along with this the response to antibiotic treatment was tested when administered at different timepoint after infection. The findings of the current article indicate that bacterial numbers are reduced in the non treated normal population model when compared to the aged model. There is also reduced neutrophil recruitment in the aged model and early intervention with antibiotic treatment is more useful in the aged model compared to the normal model. While the model can be useful for future studies, as is the current manuscript has some issues that need to be addressed and are discussed below. 

Major:

General proof reading and grammatical check needs to be performed, specifically the introduction.

The same data is used in figure 2 and 3. Figure 3 is comparing specific modeled responses in young and aged individuals to highlight the changes in bacterial numbers and neutrophils, but these figures should be altered to not repeat data presentation.

Determining levels of cytokine production for effective modelling is not explained. The current proposal works off of previous modelling platform where the authors address the levels of MCP-1 used as a function of NF-kB activation. The levels of IL-8 and CXCL5 are not addressed besides saying they are added. How are the levels chosen and how are changes due to pneumococcal infection modeled? Surface proteins of the pneumococcus, CbpA (PspC), are known to downregulate IL-8 levels during infections so it is unclear how levels of these chemoattractants are determined for the modelling system.

Minor:

Ln 31. The sentence is worded strangely. Maybe “Pneumonia is the cause of 20% of deaths in children worldwide with infection of lung alveolar tissues by Streptococcus pneumoniae being the most common cause.”

Ln 57. “bacterial infection” as opposed to “bacteria infection”

Ln 59. Add “high-“ before the word “risk”

Ln 60. Add “high-“ before the word “risk”

Ln 62. “inflammatory” instead of “inflammation”

Ln 76. Italicize “S. pneumoniae” in this sentence

Ln. 91-92. The second portion of the sentence needs to be modified for clarity. Possibly “which precedes the adaptive response by one or more days.”

Ln. 99. “primarily”

Ln 201. “”it has been observed that there is a reduction in neutrophil load in aged mice”

Ln 201. “Here,”

Ln 202. “pneumococcal”

Figure 1A. I would make the large blue arrow less opaque. It is difficult to see the LPS gradient. (Why LPS, pneumo has no LPS?)

There are variations in italicizing the latin words “in vivo”, “in vitro”, and “in silico”. Some are italicized and some are not, keep formatting consistent.

Figure 6. Indicate panel A and B

It is extremely difficult to follow some of the changes in the 10 simulations for some of the figures. If possible some way to more easily distinguish between simulations should be attempted, specifically monocyte and IL-8 figures. Possibly geom_smooth() type function when producing the graphs.

Author Response

Q1: General proof reading and grammatical check needs to be performed, specifically the introduction.

R1: We have checked the text for typos and grammar errors.

Q2:The same data is used in figure 2 and 3. Figure 3 is comparing specific modeled responses in young and aged individuals to highlight the changes in bacterial numbers and neutrophils, but these figures should be altered to not repeat data presentation.

R2: This is a good point and we agree that this new presentation of the results is better. We have merged and re-structured Figures 2 and 3 following the suggestion.

Q3: Determining levels of cytokine production for effective modelling is not explained. The current proposal works off of previous modelling platform where the authors address the levels of MCP-1 used as a function of NF-kB activation. The levels of IL-8 and CXCL5 are not addressed besides saying they are added. How are the levels chosen and how are changes due to pneumococcal infection modeled?

R3: Thanks a lot for this comment. The reviewer is right and we realize now that we did not provide detailed explanation about this model feature. In the revised version of the draft we have included, accordingly, an explanation in the methodology: “In the model we integrated in each epithelial, monocyte  and macrophage cell agent an NF-κB-centered intracellular network responsible for the triggering of the cytokines and chemokines secretion. In case of the IL-8 produced by macrophages and monocytes in response to bacteria, we assumed for IL-8 the same model parameter values than for MCP-1 based on the fact that both ligands show similar expression profiles in relevant in vitro experiments26. Further, in the model we assumed that the CXCL5 secretion is triggered in the epithelial cells through the same NF-κB-centered network that triggers MCP-1. Finally, we calculated the diffusion constants for each cytokine and chemokine in the alveolus using an empirical estimate based on the molecular weight of macromolecules27”.

Q4: Surface proteins of the pneumococcus, CbpA (PspC), are known to downregulate IL-8 levels during infections so it is unclear how levels of these chemoattractants are determined for the modelling system.

R4: This is also an interesting feature that was not addressed in the previous draft. Since we are investigating the very early phase of infection, in our model we assumed that bacteria are still adapting to the new environment. One of the adaptations that S. pneumoniae develop during this phase is the triggering of  capsule proteins synthesis. However, in our previous publication our results suggest that in early infection capsule production is low as a mechanism to favor the establishment of the bacteria population in the alveolus (see reference number 24). Under these circumstances, we expect that any immunomodulatory effects coming from S. pneumoniae is negligible at this very early phase of infection and will most likely be promoted during a subsequent phase when bacteria is established and adapted to the alveolar tissue and bacteremia is higher. In any case, we find this observation from the reviewer relevant and we have added the following text in the methodology: An important feature of S. Pneumoniae lung infection is the existence of immunomodulatory effects promoted by proteins and other compounds of the bacteria surface and capsule29–31. In our model, we assume that these immunomodulatory processes are negligible during the very initial phase of the infection here modelled, in which bacteria are adapting to the new environment and the rate of capsule proteins and compounds synthesis is low (see discussion about this issue in Santos et al. 2018)24”.

Minor comments

Ln 31. The sentence is worded strangely. Maybe “Pneumonia is the cause of 20% of deaths in children worldwide with infection of lung alveolar tissues by Streptococcus pneumoniae being the most common cause.”

Ln 57. “bacterial infection” as opposed to “bacteria infection”

Ln 59. Add “high-“ before the word “risk”

Ln 60. Add “high-“ before the word “risk”

Ln 62. “inflammatory” instead of “inflammation”

Ln 76. Italicize “S. pneumoniae” in this sentence

Ln. 91-92. The second portion of the sentence needs to be modified for clarity. Possibly “which precedes the adaptive response by one or more days.”

Ln. 99. “primarily”

Ln 201. “”it has been observed that there is a reduction in neutrophil load in aged mice”

Ln 201. “Here,”

Ln 202. “pneumococcal”

R: Thanks a lot for all the minor suggestions. We have implemented all of them and checked for additional minor typos.

Qm1:Figure 1A. I would make the large blue arrow less opaque. It is difficult to see the LPS gradient. (Why LPS, pneumo has no LPS?)

Rm1: This LPS is a mistake from previous figures. We have fixed that. Thanks for pointing out that.

Qm2: There are variations in italicizing the latin words “in vivo”, “in vitro”, and “in silico”. Some are italicized and some are not, keep formatting consistent.

Rm2:We have checked for that and fixed it correspondingly.

Qm3: Figure 6. Indicate panel A and B

Rm3: These labels were added according to the suggestion.

Qm4: It is extremely difficult to follow some of the changes in the 10 simulations for some of the figures. If possible some way to more easily distinguish between simulations should be attempted, specifically monocyte and IL-8 figures. Possibly geom_smooth() type function when producing the graphs.

Rm4: We agree on this point. We have improved the figures by playing with the line width. We consider that the visibility of the IL-8 and monocytes figures is more legible now.

Reviewer 2 Report

The manuscript ID 992237 described by Guido Santos and Julio Vera on the role of age, neutrophil infiltration and antibiotics timing in the severity of Streptococcus pneumonia. Insights from a multi-level mathematical model approach. By used a multi-level mathematical model, authors investigate the bacteria, age, antibiotics timing effect on the monocytes and neutrophils.

My comments:

The mathematical formula is not streptococcus specificity. It might be applicable in other bacterial infection if the bacteria has the similar replication cycles. To this point, I would suggest authors to indicate the application of mathematical formula.

The results “providing antibiotics to elderly patients 8 hours earlier than young ages restores in aged individuals the effective response” is interesting. However, aged patients in clinical usually stay in hospital a very long period that more than 8 hours. Thus, aged patients with multiple diseases are not suitable for such mathematical calculation.

We all knew that an early application of correct antibiotics plays a significant role in the infectious disease progression. To identify what bacteria causes the disease is time consuming.  

Minor problem:

Labels for 6A or 6B lack in figure 6.

Author Response

Major comments

Q1: The mathematical formula is not streptococcus specificity. It might be applicable in other bacterial infection if the bacteria has the similar replication cycles. To this point, I would suggest authors to indicate the application of mathematical formula.

R1: Thanks for this interesting comment. We agree that there are other reasonably prevalent bacterial species that infect lungs and behave in a way similar to S. Pneumoniae, for which our model could also simulate their infection dynamics. According to this suggestion, we have included some text in the discussion covering this possibility: Our model has been calibrated using data from experiments with S. Pneumoniae. We have focused on S. Pneumoniae because it is the most prevalent bacterial lung infection. However, the current implementation of our model could be used with little changes to simulate the infection dynamics associated to similar bacterial species”.

We have also included a sentence in the “Model description” part of Methods arguing that the model is built on top of the previous published model that was calibrated using experiments with S. pneumoniae: This model was calibrated using data from S. pneumoniae experiments in vitro and in vivo”.

Q2: The results “providing antibiotics to elderly patients 8 hours earlier than young ages restores in aged individuals the effective response” is interesting. However, aged patients in clinical usually stay in hospital a very long period that more than 8 hours. Thus, aged patients with multiple diseases are not suitable for such mathematical calculation.

R2: We consider this comment very interesting and we have modified this part of the text based on this suggestion. In essence, our proposal is to develop a protocol for protecting aged patients from pneumonia in hospitals, which would be applied when they enter the hospital for planned or unplanned intervention. The idea is to decide on applying prophylactic antibiotics based on their age, immunological conditions and neutrophils count. This proposal is shown in Figure 5 (beware of change in figure indexes). The protocol requires estimating neutrophils counts, which can be done quickly with standard blood analysis, but it does not require to make a time consuming identification of the bacterial species because we propose to use as profilactics wide-spectrum antibiotics covering the most prevalent species. Based on this clariffication, we have added the following text in the discussion: The proposed protocol would be applied as follows: patients with a planned (surgery, dyalisis, chemotherapy) or unplanned (trauma) hospitalization involving a pneumonia high-risk intervention (intubation, traqueoscopia, etc.) would be splitted into high-risk and low-risk patients based on the age and other immunological conditions. Neutrophils levels would be quantified in high-risk patients to further determine the pneumonia risk. Finally, patients categorized in high-risk based on both age/immune state and low neutrophil counts would receive prophylactic antibiotics in order to reduce their risk of bacterial pneumonia during or post-intervention”.

Minor comments

Q: Labels for 6A or 6B lack in figure 6.

R: These labels were added according to the suggestion.